

# Phylogenetic relationships in the southern African genus *Drosanthemum* (Ruschioideae, Aizoaceae)

Sigrid Liede-Schumann[1], Guido W. Grimm[2], Nicolai M. Nürk[1], Alastair J. Potts[3], Ulrich Meve[1] and Heidrun E.K. Hartmann[4],[†]

[1] Department of Plant Systematics, University of Bayreuth, Bayreuth, Germany
[2] Unaffiliated, Orléans, France
[3] African Centre for Coastal Palaeoscience, Nelson Mandela University, Port Elizabeth, Eastern Cape, South Africa
[4] Department of Systematics and Evolution of Plants, University of Hamburg, Hamburg, Germany
[†] Deceased.

Corresponding author
Sigrid Liede-Schumann,
sigrid.liede@uni-bayreuth.de

## ABSTRACT

**Background**. *Drosanthemum*, the only genus of the tribe Drosanthemeae, is widespread over the Greater Cape Floristic Region in southern Africa. With 114 recognized species, *Drosanthemum*, together with the highly succulent and species-rich tribe Ruschieae, constitute the 'core ruschioids' in Aizoaceae. Within *Drosanthemum*, nine subgenera have been described based on flower and fruit morphology. Their phylogenetic relationships, however, have not yet been investigated, hampering understanding of monophyletic entities and patterns of geographic distribution.

**Methods**. Using chloroplast and nuclear DNA sequence data, we performed network- and tree-based phylogenetic analyses of 73 species of *Drosanthemum* with multiple accessions for widespread species. A well-curated, geo-referenced occurrence dataset comprising the 134 genetically analysed and 863 further accessions was used to describe the distributional ranges of intrageneric lineages and the genus as a whole.

**Results**. Phylogenetic inference supports nine clades within *Drosanthemum*, seven of which group in two major clades, while the remaining two show ambiguous affinities. The nine clades are generally congruent to previously described subgenera within *Drosanthemum*, with exceptions such as cryptic species. In-depth analyses of sequence patterns in each gene region were used to reveal phylogenetic affinities inside the retrieved clades in more detail. We observe a complex distribution pattern including widespread, species-rich clades expanding into arid habitats of the interior (subgenera *Drosanthemum* p.p., *Vespertina, Xamera*) that are genetically and morphologically diverse. In contrast, less species-rich, genetically less divergent, and morphologically unique lineages are restricted to the central Cape region and more mesic conditions (*Decidua, Necopina, Ossicula, Quastea, Quadrata, Speciosa*). Our results suggest that the main lineages arose from an initial rapid radiation, with subsequent diversification in some clades.

## INTRODUCTION

In the south-western corner of Africa, the iconic leaf-succulent Aizoaceae (ice plant family, including *Lithops*, 'living stones'; Caryophyllales) is one of the most species-rich families in the biodiversity hot-spot of the Greater Cape Floristic Region (GCFR; *Born, Linder & Desmet, 2007*; *Mittermeier et al., 1998*; *Mittermeier et al., 2004*; *Mittermeier et al., 2011*), ranking second in the number of endemic genera and fifth in the number of species (*Manning & Goldblatt, 2012*). Although Aizoaceae species have received much attention both in terms of their ecology and evolution (e.g., *Klak, Reeves & Hedderson, 2004*; *Valente et al., 2014*; *Ellis, Weis & Gaut, 2007*; *Hartmann, 2006*; *Schmiedel & Jürgens, 2004*; *Powell et al., 2019*), information on phylogenetic relationships within major clades (or subfamilies) is still far from complete. Here, we aim at filling some of the knowledge-gaps by: (1) providing a review of the current classification of the family, and origin and distribution of major clades (in the Introduction), and (2) a study of phylogenetic relationships in the enigmatic and hitherto, phylogenetically, almost neglected genus *Drosanthemum*.

### Subfamilies of Aizoaceae: relationship of major clades

Aizoaceae currently comprises ca. 1800 species (*Hartmann, 2017a*; *Klak, Hanáček & Bruyns, 2017a*) classified in 145 genera and five subfamilies (*Klak, Hanáček & Bruyns, 2017a*). The first three subfamilies—Sesuvioideae, Aizooideae, Acrosanthoideae—are successive sisters to Mesembryanthemoideae + Ruschioideae (*Klak et al., 2003*; *Klak, Reeves & Hedderson, 2004*; *Thiede, 2004*; *Klak, Hanáček & Bruyns, 2017b*; for authors and species numbers see Table 1). Species of Mesembryanthemoideae and Ruschioideae, commonly referred to as 'mesembs' (Mesembryanthema; *Hartmann, 1991*), were found in molecular phylogenetic studies to be reciprocally monophyletic (e.g., *Klak et al., 2003*; *Thiede, 2004*; *Klak, Hanáček & Bruyns, 2017b*).

Mesembryanthemoideae and Ruschioideae, as well as their sister-group relationship, are supported by morphological characters. Mesembryanthemoideae + Ruschioideae can be distinguished from the remaining Aizoaceae by raphid bundles of calcium oxalate (in contrast to calcium oxalate druses), the presence of petals of staminodial origin, half-inferior or inferior ovary and a base chromosome number of $x = 9$ (*Bittrich & Struck, 1989*). The conspicuous loculicidal hygrochastic fruit capsules of ca. 98% of the species (*Ihlenfeldt, 1971*; *Parolin, 2001*; *Parolin, 2006*) are lacking in Acrosanthoideae, for which xerochastic, parchment-like capsules are apomorphic, but are also predominant in subfamily Aizooideae (*Bittrich, 1990*; *Klak, Hanáček & Bruyns, 2017a*). In Aizooideae, however, valve wings of the capsules are either absent or very narrow, while they are well developed in Mesembryanthemoideae + Ruschioideae (*Bittrich & Struck, 1989*).

The capsules of Mesembryanthemoideae and Ruschioideae differ in the structure of their expanding keels. The expanding keels are of purely septal origin in Mesembryanthemoideae, and mainly of valvar origin in Ruschioideae (*Hartmann, 1991*). In floral structure, Ruschioideae are characterized almost always by crest-shaped (lophomorphic) nectaries and a parietal placentation (*Hartmann & Niesler, 2009*), while Mesembryanthemoideae possess plain shell-shaped (coilomorphic) nectaries and a central placentation.

**Table 1 Infrafamilial classification of Aizoaceae.**

| | Subfamilies | Tribes |
|---|---|---|
| | **Sesuvioideae** Lindl. | |
| | **Aizooideae** Arn. | |
| | **Acrosanthoideae** Klak | |
| **"Mesembs"** | **Mesembryanthemoideae** Ihlenf., Schwantes & Straka | |
| | **Ruschioideae** Schwantes | **Apatesieae** Schwantes ex Ihlenf., Schwantes & Straka |
| | | **Dorotheantheae** (Schwantes ex Ihlenf. & Struck) Chess., Gideon F.Sm. & A.E.van Wyk |
| | | **"core ruschioids"** — **Drosanthemeae** Chess., Gideon F.Sm. & A.E.van Wyk |
| | | **Ruschieae** Schwantes ex Ihlenf., Schwantes & Straka |

## Subfamilies of Aizoaceae: origin and distribution of major clades

Subfamily Sesuvioideae, sister to the rest of Aizoaceae, originated in Africa/Arabia suggesting an African origin for the entire family (*Bohley et al., 2015*). While Sesuvioideae and Aizooideae dispersed as far as Australia and the Americas (*Bohley et al., 2015*; *Klak, Hanáček & Bruyns, 2017b*), Acrosanthoideae, Mesembryanthemoideae and Ruschioideae are most diverse in southern Africa. Only a small number of Ruschioideae species are found outside of this area. *Delosperma* N.E.Br. is native to Madagascar and Réunion and expands with less than ten species along the East African mountains into the south-eastern part of the Arabian Peninsula (*Hartmann, 2016*; *Liede-Schumann & Newton, 2018*). Additionally, in Ruschioideae there are nine halophytic species endemic to Australia (*Prescott & Venning, 1984*; *Hartmann, 2017a*; *Hartmann, 2017b*), and possibly one species to Chile (*Hartmann, 2017a*).

In southern Africa, most species of Acrosanthoideae, Mesembryanthemoideae and Ruschioideae are native to the Winter Rainfall Region (*Verboom et al., 2009*; *Valente et al., 2014*) in the GCFR. Acrosanthoideae with only six species is endemic to mesic fynbos, whereas Mesembryanthemoideae and Ruschioideae are speciose in more arid Succulent Karoo vegetation (*Klak, Hanáček & Bruyns, 2017a*).

## Within Ruschioideae: relationships of major clades

Ruschioideae constitute the largest clade of Aizoaceae with estimated species richness of ca. 1,600 (*Stevens, 2001*, onwards; *Klak, Bruyns & Hanáček, 2013*). Within Ruschioideae three tribes, Apatesieae, Dorotheantheae, and Ruschieae s.l., have been distinguished based on unique combinations of nectary and capsule characters (*Chesselet, Smith & Van Wyk, 2002*). These three tribes form well supported clades in phylogenetic analyses (*Klak et al., 2003*; *Thiede, 2004*; *Valente et al., 2014*). Ruschieae s.l. are further characterized by the possession of wideband tracheids (*Landrum, 2001*), endoscopic peripheral vascular bundles in the leaves (*Melo-de Pinna et al., 2014*), smooth and crested mero- and holonectaries, well-developed valvar expanding tissue in the capsules (*Hartmann & Niesler, 2009*), the loss of the *rpoC1* intron in the chloroplast DNA (cpDNA; *Thiede, Schmidt & Rudolph, 2007*), and the possession of two *ARP* (Asymmetric Leaves1/Rough Sheath 2/Phantastica) orthologues in the nuclear DNA (*Illing et al., 2009*); the duplication most likely took place after the divergence of the Ruschioideae from the Mesembryanthemoideae, with the subsequent loss of one paralogue in Apatesieae and Dorotheantheae (*Illing et al., 2009*).

Within Ruschieae s.l. ('core ruschioids' *sensu Klak, Reeves & Hedderson, 2004*), *Klak et al. (2003)* additionally revealed two clades with strong support, Ruschieae s.str. and a clade consisting only of members of *Drosanthemum* Schwantes. Species of *Delosperma*, considered closely related to *Drosanthemum* due to a papillate epidermis, often broad, flat mesophytic leaves, relatively simple hygrochastic fruits and a meronectarium have been described with *Drosanthemum* in tribe Delospermeae Chess., Gideon F.Sm. & A.E.van Wyk (*Chesselet, Smith & Van Wyk, 2002*). In phylogenetic studies, however, *Delosperma* species are nested in Ruschieae s.str. (except for a few species, e.g., *Drosanthemum asperulum* and *D. longipes*, which have been assigned in turn to either *Delosperma* or *Drosanthemum*). Consequently, *Chesselet, Smith & Van Wyk (2004)* included *Delosperma* in Ruschieae s.str. and coined the monogeneric Drosanthemeae Chess., Gideon F. Sm. & A.E. van Wyk as a distinct tribe sister to Ruschieae s.str. (in the following Drosanthemeae + Ruschieae = core ruschioids; Table 1).

While Ruschieae are characterized by fused leaf bases (*Chesselet, Smith & Van Wyk, 2002*), an apomorphic trait is less obvious for its sister tribe Drosanthemeae. *Hartmann & Bruckmann (2000)* suggested capsules with a bipartite pedicel, of which the lower part appears darker due to an inner corky layer, and the upper part often thinner and agreeing in surface and colour with the capsule base. More generally, species of Drosanthemeae are considered mesomorphic, compared to the highly succulent, xeromorphic Ruschieae (*Klak, Bruyns & Hanáček, 2013*).

## Core ruschioids: relationships of lineages

A sister-group relationship of Drosanthemeae and Ruschieae has been revealed by molecular phylogenetic analyses (*Klak, Bruyns & Hanáček, 2013*). Whether both groups are reciprocally monophyletic (and in which circumscription) is less clear (e.g., *Klak, Hanáček & Bruyns, 2017b*). For example, molecular phylogenies identified two species erroneously included in Drosanthemeae. One of these, *Drosanthemum diversifolium* L.Bolus, was first transferred to *Knersia* H.E.K.Hartmann & Liede, a monotypic genus placed in Ruschieae

(*Hartmann & Liede-Schumann, 2013*), and later to *Drosanthemopsis* Rauschert (Ruschieae) by *Klak, Hanáček & Bruyns (2018)*. The second species, *Drosanthemum pulverulentum* (Haw.) Schwantes, with a xeromorphic epidermis untypical for Drosanthemeae, was retrieved as member of the highly succulent clade "L1" in Ruschieae (*Klak, Bruyns & Hanáček, 2013*; not yet formally transferred). With these corrections, Drosanthemeae comprise a single genus, *Drosanthemum*, with 114 species presently recognized and a wide distribution in the GCFR with the centre of diversity in the Cape Floristic Region (*Hartmann, 2017a*; *Van Jaarsveld, 2015*; *Van Jaarsveld, 2018*; *Liede-Schumann, Meve & Grimm, 2019*).

### *Drosanthemum* (Drosanthemeae) systematics

Within *Drosanthemum*, five floral types have been distinguished, differing mainly in number, position and relative length of petaloid staminodes (*Rust, Bruckmann & Hartmann, 2002*; Fig. 1). Also, ten types of capsules have been described, differing in size and shape of the capsule base and the capsule membrane, and the presence or absence of a closing body (*Hartmann & Bruckmann, 2000*). Based on a combination of these flower and fruit types, *Hartmann (2007)* proposed a subdivision of *Drosanthemum* in eight subgenera. Later, *Hartmann & Liede-Schumann (2014)* proposed two more subgenera based on additional vegetative morphology, and also suggested the union of two of the previously described subgenera. This reflects an unusually broad variation in flower and capsule types encountered in the genus compared with other Aizoaceae genera.

Despite this extraordinary morphological diversity, molecular phylogenetic studies of *Drosanthemum* have hitherto been restricted to few species: nine species studied for ten cpDNA regions in *Klak, Bruyns & Hanáček (2013)* and 16 species studied for two cpDNA regions and the nuclear-encoded internal transcribed spacer (ITS) region of the 35S ribosomal DNA cistron in *Hartmann & Liede-Schumann (2013)*. Obtaining increased species coverage representative for the phenotypic and taxonomic diversity present in *Drosanthemum* is challenging partly due to ambiguous species assignment to either *Drosanthemum* or *Delosperma* (*Hartmann & Liede-Schumann, 2014*), but mainly due to challenges in attributing specimens to published species names in *Drosanthemum*. Ambiguous and/or overlapping diagnostic characters are common among closely related species and also present between subgenera or genera. Specimens of species flocks and cryptic species (*Liede-Schumann, Meve & Grimm, 2019*) are often hard to identify with certainty, a fact that might have hampered investigation of the genetic differentiation among *Drosanthemum* species. In this study, we build on Heidrun Hartmann's huge field collections of identified specimens of *Drosanthemum*. The present study would not have been possible without her enduring commitment to collect, diagnose, and formally name species in the Aizoaceae.

We present a phylogenetic study of *Drosanthemum* covering more than 64% of the species richness (73 of 114 recognized species) representing all subgenera. We analyse chloroplast and nuclear DNA sequence variation using phylogenetic tree and network approaches and assemble a taxonomically-verified occurrence dataset. Specifically, we test whether: (1) *Drosanthemum* is a monophyletic lineage sister to Ruschieae; (2) the

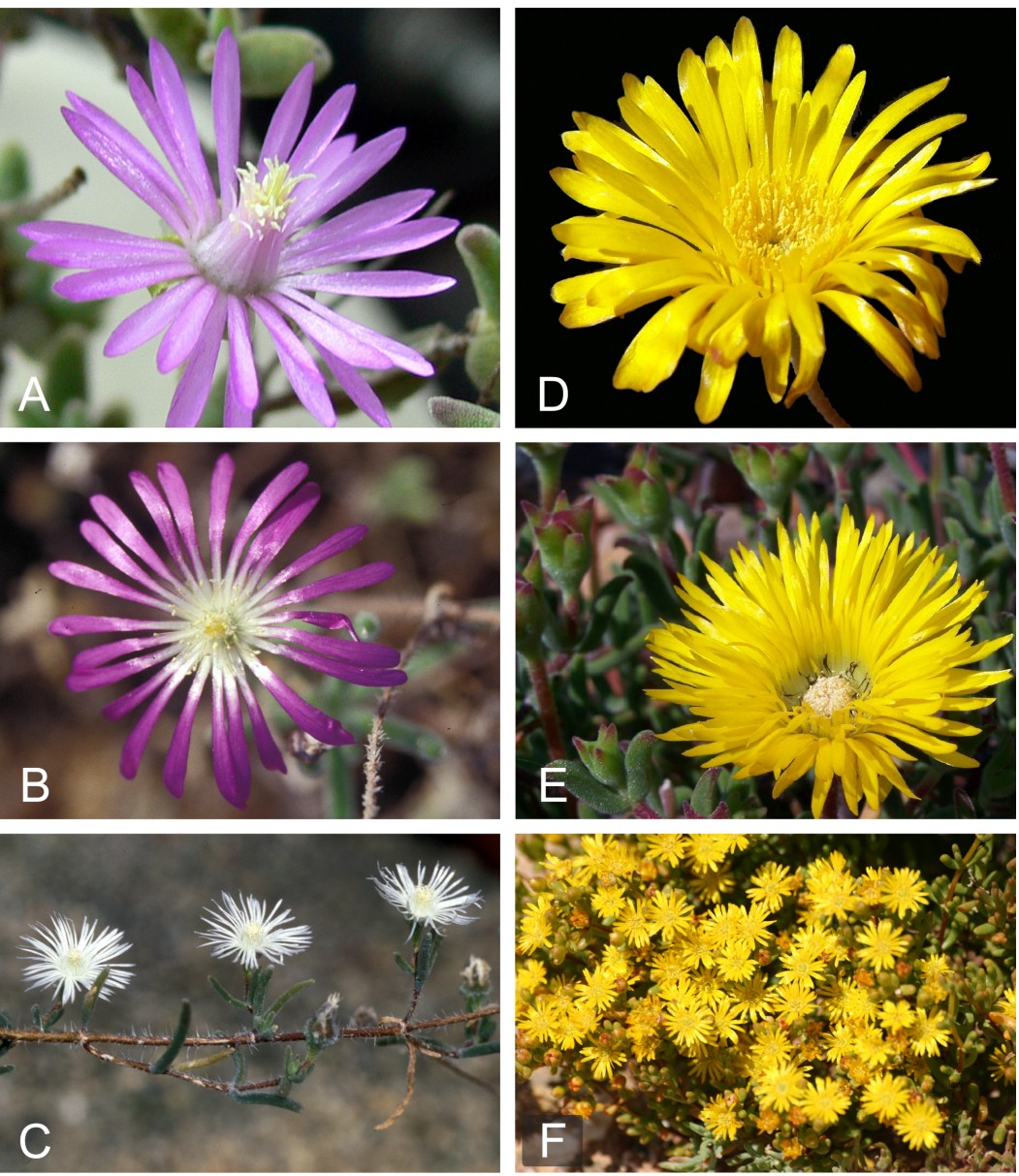

**Figure 1 Floral diversity in *Drosanthemum*.** (A) *Drosanthemum lique* (subg. *Vespertina*, clade IIIa; HH 34800); (B) *Drosanthemum nordenstamii* (subg. *Drosanthemum*, clade Ia; HH 31525); (C) *Drosanthemum papillatum* (subg. *Quastea*, clade VI; HH 32425); (D) *Drosanthemum cereale* (subg. *Speciosa*, clade Vb; HH 34489)—note the absence of black staminodes; (E) *Drosanthemum hallii* (subg. *Speciosa*; clade Vb; HH 34610)—note the black staminodes; (F) *Drosanthemum zygophylloides*. Photos (A–E): H.E.K. Hartmann; F: L. Mucina.

morphologically delineated subgenera are monophyletic, in particular, whether the most species-rich subgenus *Drosanthemum* is monophyletic or, alternatively, a "dustbin" for species that cannot be assigned to other subgenera based on morphology; (3) all accessions within currently recognized species are indeed each other's closest relatives; and (4) the clades detected in this study have distinct geographic distributions in the GCFR.

## MATERIAL AND METHODS

### Taxon sampling

We established a collection of georeferenced and identified *Drosanthemum* samples; each sample was only included if sufficient material was available to identify key characteristics. The full collection ('core collection'; $n = 997$ samples) represents the most comprehensive sampling of the currently recognized *Drosanthemum* species, covering 85 species in total, with each species represented by up to 30 georeferenced samples (range: 1–30; mean: 5 samples per species). This core collection consists of 590 samples identified to subgenus, plus 407 identified to species. The subgeneric classification follows *Hartmann (2007)* and *Hartmann & Liede-Schumann (2014)*.

A subset of the core collection was used to generate the molecular dataset; this subset comprised 134 accessions of *Drosanthemum*, covering 73 of the recognized species, with the more widespread and morphologically variable species represented by up to 5 accessions. To cover the full distribution in the larger subgenera, the molecular dataset comprises 21 accessions identified to subgenus, several of which most likely represent hitherto undescribed species: subg. *Drosanthemum* (10 accessions), subg. *Vespertina* (8 accessions), subg. *Xamera* (2 accessions), and subg. *Ossicula* (1 accession). Geographic distributions of the subgenera that were corroborated with phylogenetic inference in this study (i.e., inferred clades, see *Results*) were plotted on a map using the elevation above sea level data from the WorldClim climate layers (*Hijmans et al., 2005*), with a spatial resolution of 30′ using the RASTER library v2.8-19 (*Hijmans, 2019*) in R v3.5.3 (*R Core Team, 2019*). Geographic references for the core collection are available at the Dryad digital repository (*Liede-Schumann et al., 2019*).

For outgroup comparison we selected a broad spectrum of species representing the three remaining tribes of Ruschioideae: Apatesieae (two accessions representing one species), Dorotheantheae (three species), and Ruschieae (49 accessions representing 47 species and 42 genera). We used the cpDNA dataset of *Klak, Bruyns & Hanáček (2013)* pruned to include one to several accessions of each Ruschieae clade (depending on clade size) with an additional nine species sequenced in previous studies of the present authors. Nuclear ITS sequences were downloaded from GenBank for accessions identical to the cpDNA dataset; in five cases different accessions of the same species had to be used: *Dorotheanthus bellidiformis* (Burm.f.) N.E.Br., *Cheiridopsis excavata* L.Bolus, *Corpuscularia lehmannii* (Eckl. & Zeyh.) Schwantes, *Jacobsenia kolbei* (L.Bolus) L.Bolus & Schwantes, and *Prepodesma orpenii* (N.E.Br.) N.E.Br. A species shown by *Klak, Bruyns & Hanáček (2013)*, to belong in Ruschieae clade L1, *Drosanthemum pulverulentum* (Haw.) Schwantes, was regarded as part of the outgroup.

### PCR and sequencing

We targeted four cpDNA markers and the nuclear rDNA ITS region. These included two cpDNA markers, the *trn*S-*trn* G intergenic spacer region and the *rpl* 16 intron, that were found to have the highest intra-generic divergence amongst the seven *Drosanthemum* accessions used by *Klak, Bruyns & Hanáček (2013)*; these regions were amplified using the primers and protocols provided in the original paper. In addition, two cpDNA intergenic

spacers, *trnQ–5′rps16* and *3′rpS16–5′trn*K, were amplified with primers trnQ[(UUG)] and rpS16x1 and with primers rpS16x2F2 and trnK[(UUU)], respectively (*Shaw et al., 2007*). The nuclear ITS region was amplified as detailed in *Hassan, Thiede & Liede-Schumann (2005)*.

Total genomic DNA was extracted from seedlings or from herbarium specimens using the DNeasy Plant MiniKit (Qiagen, Hilden, Germany), following the protocol of the manufacturer. For sequencing, the PCR products were sent to Entelechon (Regensburg, Germany) or Eurofins (Ebersberg, Germany) resulting in 473 new sequences of *Drosanthemum* species produced in this study. Forward and reverse sequences were aligned with CODONCODE ALIGNER, v.3.0.3 (CodonCode Corp., Dedham, Massachusetts, USA). Sequence data of individual marker regions were aligned with OPAL (*Wheeler & Kececioglu, 2007*) and checked visually using MESQUITE v.3.51 (*Maddison & Maddison, 2018*). All sequences newly generated in this study have been submitted to ENA (for accession numbers see Supplemental Information 1).

## Phylogenetic analyses
### Phylogenetic tree inference
We used maximum likelihood (ML) and non-parametric bootstrapping (BS) analysis on a concatenated cpDNA dataset (comprising all four regions) including only *Drosanthemum* species ('*Drosanthemum*' dataset: 134 accessions), and on a dataset also including outgroup species ('Ruschioideae' dataset: 188 accessions; see *Taxon sampling*) to infer the placement of the *Drosanthemum* species in relation to the other Ruschioideae lineages. Note that prior to this concatenated cpDNA analysis, each marker was analysed individually and in various division schemes (several data matrices were tested: partitioned and unpartitioned, also including or excluding the most-divergent and length-polymorphic *rps16-trn*Q spacer region, and including/excluding an ITS partition; raw data, code and results are available at Dryad, *Liede-Schumann et al., 2019*). No supported topological discordances were present; thus we used a concatenated four-markers cpDNA dataset. ML tree inference and BS analysis relied on RAxML v. 8.0.20 (*Stamatakis, 2014*), partitioned and set to allow for site-specific variation modelled using the 'per-site rate' model approximation of the Gamma distribution (*Stamatakis, 2006*). Duplicate sequences were reduced to a single sequence resulting in 131 accessions in the ML cpDNA tree of *Drosanthemum*. The same RAxML settings were used for the 'Ruschioideae' dataset. To obtain probability estimates for the most likely *Drosanthemum* (ingroup) root, we used the evolutionary placement algorithm (EPA; *Berger, Krompass & Stamatakis, 2011*) implemented in RAxML and following the analytical set-up of *Hubert et al. (2014)* and *Grímsson, Grimm & Zetter (2018)*. EPA provides probability estimates (*Berger, Krompass & Stamatakis, 2011*) for placing a query sequence (here: outgroup taxa representing the Ruschieae) within a given topology (here: ML *Drosanthemum* cpDNA tree) offering identifying a consensus outgroup-based root while minimising potential biases (e.g., long-branch attraction, LBA; *Bergsten, 2005*). To do so, we queried a set of 47 Ruschieae species and calculated a probability estimate ($p_R$) by averaging the likelihood weight ratios of query taxa per inferred rooting scenario over all queried taxa.

### Phylogenetic network inference

We investigated competing support patterns within *Drosanthemum* by means of BS consensus networks (*Holland & Moulton, 2003*; *Grimm et al., 2006*; *Schliep et al., 2017*) using SplitsTree v. 4.1.13 (*Huson & Bryant, 2006*) and up to 1000 BS (pseudo-) replicate RAxML trees (see paragraph above). The number of necessary BS replicates was determined using the extended majority bootstrap criterion (*Pattengale et al., 2009*). Additionally, we investigated within-lineage differentiation of subclades within *Drosanthemum* ('subclade' refers here to the nine clades within the genus *Drosanthemum* defined in the Results section) using median-joining (MJ; *Bandelt, Forster & Röhl, 1999*) networks for the cpDNA dataset and statistical parsimony (SP; *Templeton, Crandall & Sing, 1992*) networks for the ITS data. MJ networks were computed with Network v.5.0.0.3 (Fluxus; available online http://www.fluxus-engineering.com/sharenet.htm) with default settings and no character weighting and SP networks with pegas v0.11 (*Paradis, 2010*) in R. In the MJ network analyses, we used reduced sequence alignments differentiating four sequence patterns at the intra-subclade level: (i) single-nucleotide polymorphisms (SNPs); (ii) insertions, duplications and deletions (indels), represented by a single character because gaps are treated as 5th base by Network by default; (iii) length-polymorphic sequence motifs (LP, such as multi-A motifs, which were only considered when including mutations additional to length variation; this category also includes more complex length-polymorphic patterns such as length-polymorphic AT-dominated sequence regions); and (iv) oligo-nucleotide motifs (ONM), short motifs with apparently linked mutations that can slightly differ in length, which were treated as a single mutational event; inversions, like the ones found in the pseudo-hairpin structure of the *trnK-rps16* spacer, are a special form of ONMs. The highly divergent, length-polymorphic 'high-div' region characterising the 5′ end of the *rps* 16-*trn*Q intergenic spacer, was generally excluded from the analysis but included in the haplotype documentation (see *Liede-Schumann et al., 2019*: file Haplotyping.xlsx).

The reasoning for the use of MJ and SP networks is because there were few consistent mutations at the intrageneric level within subclades—this results in a flat likelihood surface of the tree space and, in this situation, parsimony can be more informative than probabilistic approaches (*Felsenstein, 2004*). In contrast to phylogenetic trees, MJ networks include all equally parsimonious solutions to a dataset and produce *n*-dimensional splits graphs that can include topological alternatives. Also, MJ and SP haplotype networks directly depict ancestor-descendant relationships, and hence, can assist in deciding whether inferred clades in the tree are monophyletic in a strict sense, i.e., groups of inclusive common origin (*Hennig, 1950*; see also *Felsenstein, 2004*, chapter 10). Because the MJ networks can easily become diffuse or complex, especially when analysing interspecific relations, we summarized the inferred haplotypes into haplotype groups for visualizing and interpreting MJ networks.

**Table 2  Alignment and analysis parameters for the targeted sequence regions.**

| Gene region | Matrix dimension (OTU × characters) | NLD | PUC | DAP | NBS | Approx. model |
|---|---|---|---|---|---|---|
| ITS | 112 × 440[a] | 27 | 2.7% | 130 | 500 | aabcde |
| *trnS-trnG* | 121 × 1157 | 34 | 30.8% | 305 | 800 | aabbba |
| *rpl16* intron | 112 × 1193 | 36 | 22.0% | 201 | 450 | aabaaa |
| *trnK-rps16* | 122 × 768 | 31 | 20.6% | 241 | 450 | aabccc |
| *rps16-trnQ* | 127 × 788[b] | 19 | 21.8% | 245 | 550 | aababa |

Notes.
[a] Only ITS1 and ITS2, flanking rRNA and 5.8S rRNA genes not included.
[b] After exclusion of the 'high-div' region.
NLD, number of literally duplicate (identical) sequences; PUC, proportion of undetermined matrix cells (gappyness); DAP, number of distinct alignment patterns; NBS, number of necessary BS pseudoreplicates; Approx. model, approximate of the DNA substitution model optimized by RAxML for each gene region (in alphabetical order: A ↔ C, A ↔ G, A ↔ T, C ↔ G, C ↔ T, G ↔ T).

# RESULTS

## Patterns of DNA sequence diversity

We targeted the most variable cpDNA gene regions currently known for Aizoaceae, which provided a relatively high number of distinct alignment patterns (Table 2), although each cpDNA marker on its own provides low topological resolution (single plastid gene-region ML trees and BS consensus networks are provided in *Liede-Schumann et al. (2019)*. Length-polymorphism was common, hence, the high proportion of gaps (undetermined cells) in the alignments, but often restricted to duplications or deletions, rarely insertions, and explicitly alignable. An exception was the *rps*16-*trn*Q intergenic spacer, which includes regions with extreme length-polymorphism and highly complex sequence patterns that are only alignable amongst closely related species. A notable feature is a 'pseudo-hairpin' sequence found in the *trn*K-*rps*16 intergenic spacer, which includes a partly clade-diagnostic strictly complementary upstream-downstream sequence pattern composed of duplications of two short sequence motifs and subsequent deletions and a ''terminal'' inversion (shown in the coding example in Supplemental Information 2, Fig. S2A; for more details see *Liede-Schumann et al., 2019*: Haplotype.xlsx).

In general, cpDNA sequence patterns in *Drosanthemum* are highly diagnostic at and below the level of major clades, in most cases allowing identification of haplotypes or clade-unique substitution pattern. This includes a few, potentially synapomorphic (*sensu Hennig, 1950*: uniquely shared derived traits) single-base mutations in generally length-homogenous sequence portions (see *Liede-Schumann et al., 2019*). Indel patterns appear to be largely homoplastic, but sometimes diagnostic at the species level or for species flocks. In contrast, mutation patterns in the length-homogeneous (SNPs) and length-polymorphic regions (LP, indels, ONMs) are largely congruent, with few conflicting signals, for taxon splits.

The nuclear-encoded ITS region has low divergence and contains little signal for tree discrimination, which is typical for the Aizoaceae (e.g., *Klak, Bruyns & Hanáček, 2013*), and was not included for defining major clades or testing their coherence with the earlier proposed subgenera. Still, the genetic diversity present (Table 2) allows for the identification

of more ancestral vs. more derived genotypes (Supplemental Information 3), which were mapped onto the cpDNA tree (Fig. 2).

## Phylogenetic inference and potential *Drosanthemum* roots

The ML tree of *Drosanthemum* (based on the concatenated cpDNA dataset) indicates nine moderately (>65% BS support) to well supported (>90% BS support) clades (Fig. 2). Seven of these group in two major clades, with high support for the clade I+II+III+IV (98% BS support; addressed informally as '*Drosanthemum* core clade', Fig. 3) and low support for the second clade V+VI+VII (58% BS support). Clades VIII and IX have ambiguous affinities (results not shown; for full documentation see *Liede-Schumann et al., 2019*). Two species, *D. longipes* (sister to clade VII) and *D. zygophylloides* (sister to VIII) are not included in the nine described clades (see *Discussion –Phylogenetic inference reflects taxonomic classification*). Notably, the nine clades overall group into six lineages (clades I–IV, V+VI, VII+ *D. longipes*, VIII, XI, and *D. zygophylloides*), but relationships among the six lineages were weakly supported. Specifically, the earliest branching events in the ML tree are ambiguously resolved (Fig. 2)

The ML tree of Ruschioideae (based on the 'Ruschioideae' dataset) inferred *Drosanthemum* a monophyletic sister to Ruschieae (100% BS support, Fig. S4.1), supporting the 'core ruschioids'' hypothesis (Table 1; for details see *Liede-Schumann et al., 2019*). In this tree, the topology within *Drosanthemum*, however, differs in parts (clade VIII and IX successive sister to the rest; not supported) from the tree inferred by the analysis of the '*Drosanthemum*' dataset (Fig. 2; both datasets comprise the same four concatenated cpDNA regions). Taken together, phylogenetic inference is consistent with a rapid initial diversification within *Drosanthemum* that was potentially too fast to leave a signal in cpDNA sequence variation in the studied markers.

Placement of the 49 queried outgroup taxa indicates eleven potential *Drosanthemum* root positions (Fig. 3). However, six of these positions are unlikely considering the probability estimates $p_R$ (an order of magnitude lower), number of supporting queries (0 to 2), and phylogenetic evidence (Fig. S4). The remaining five root positions are summarized as follows: scenario 1, clade I–IV sister to clade V–IX, supported by 33 queries and $p_R = 0.26$ (Fig. 2; Fig. S4.2); scenario 2, clade IX sister to the rest, eight queries and $p_R = 0.23$ (Fig. S4.3); scenario 3, clade VIII + *D. zygophylloides* sister to the rest, one query and $p_R = 0.14$ (Fig. S4.4); scenario 4, cladeV–VII + *D. longipes* sister to clade I–IV + VIII + *D. zygophylloides* + IX, four queries and $p_R = 0.14$ (Fig. S4.5); scenario 5, clade I–IV + VIII + *D. zygophylloides* sister to clade V–IV + *D. longipes* + IX, supported by zero queries and $p_R = 0.14$ (Fig. S4.6). Because the outgroup samples are notably distant in the targeted plastid gene regions to *Drosanthemum* favouring attraction of most distinct accessions, scenarios 3–5 may be artefacts generated by outgroup-ingroup (long) branch attraction. Scenario 1 is identified as the most likely root position and has additionally the highest probability estimate and number of supporting queries by the distribution of ITS genotypes, indicating underived variants in clade I and V and both un- and derived ITS variants in the smaller clades outside '*Drosanthemum* core clade' (Fig. 2; see also *Results, Identification*

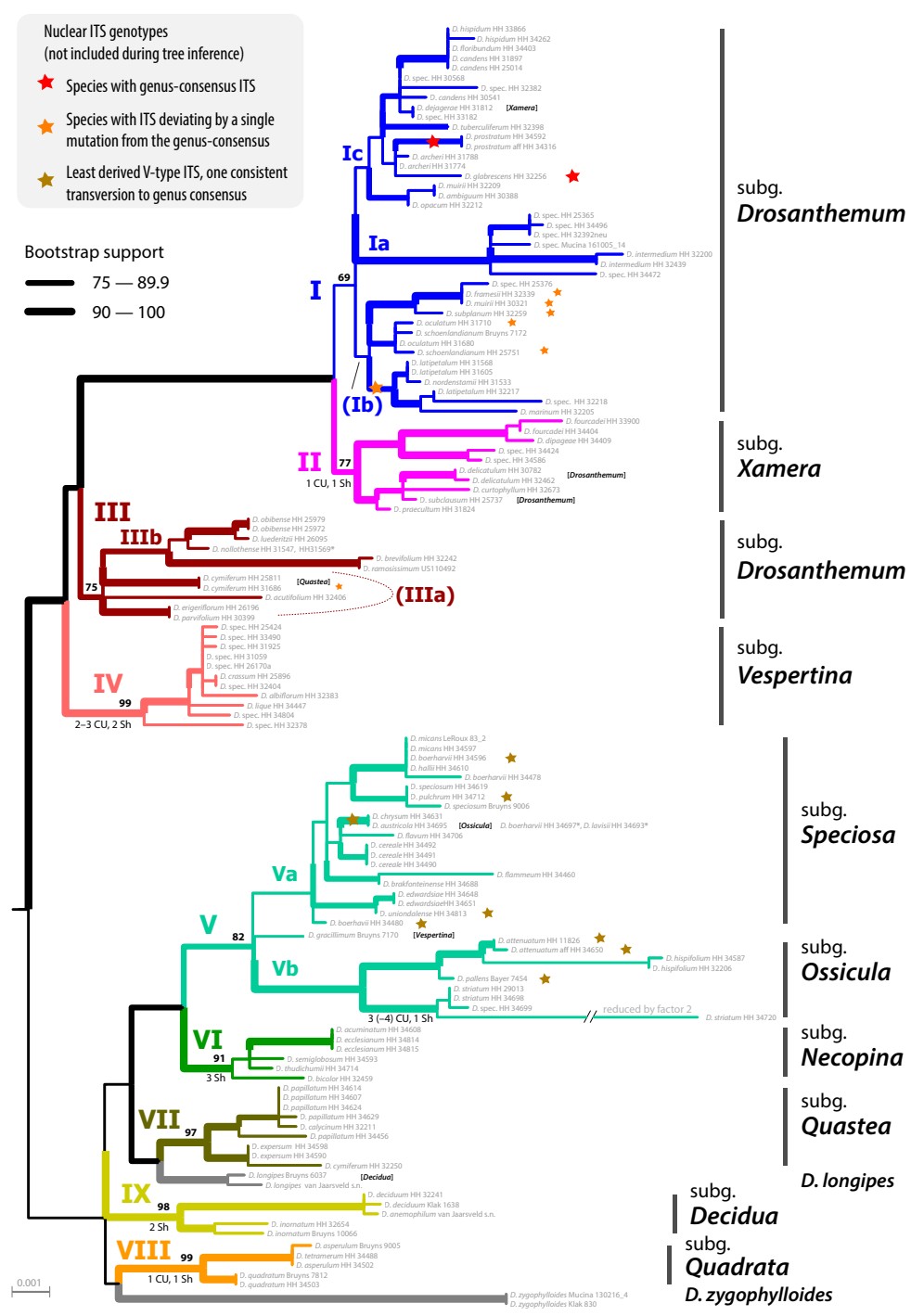

**Figure 2 Phylogeny of *Drosanthemum*.** ML tree inferred by partitioned analysis of the cpDNA sequence data. Edge lengths are scaled on expected number of substitutions. The nine main clades are annotated by roman numbers I–IX and coloured branches, with ML bootstrap support indicated by edge width (values given for the nine main clades). Bars and names to the right indicate subgeneric classification *sensu Hartmann, 2007*. An asterisk after tip names indicate accessions with literally duplicate sequences. CU, clade unique ITS mutation pattern(s); Sh, shared ITS mutation pattern found occasionally also in other clades. Rooting is according to the most likely position inferred by outgroup-EPA (scenario 1; outgroups removed).

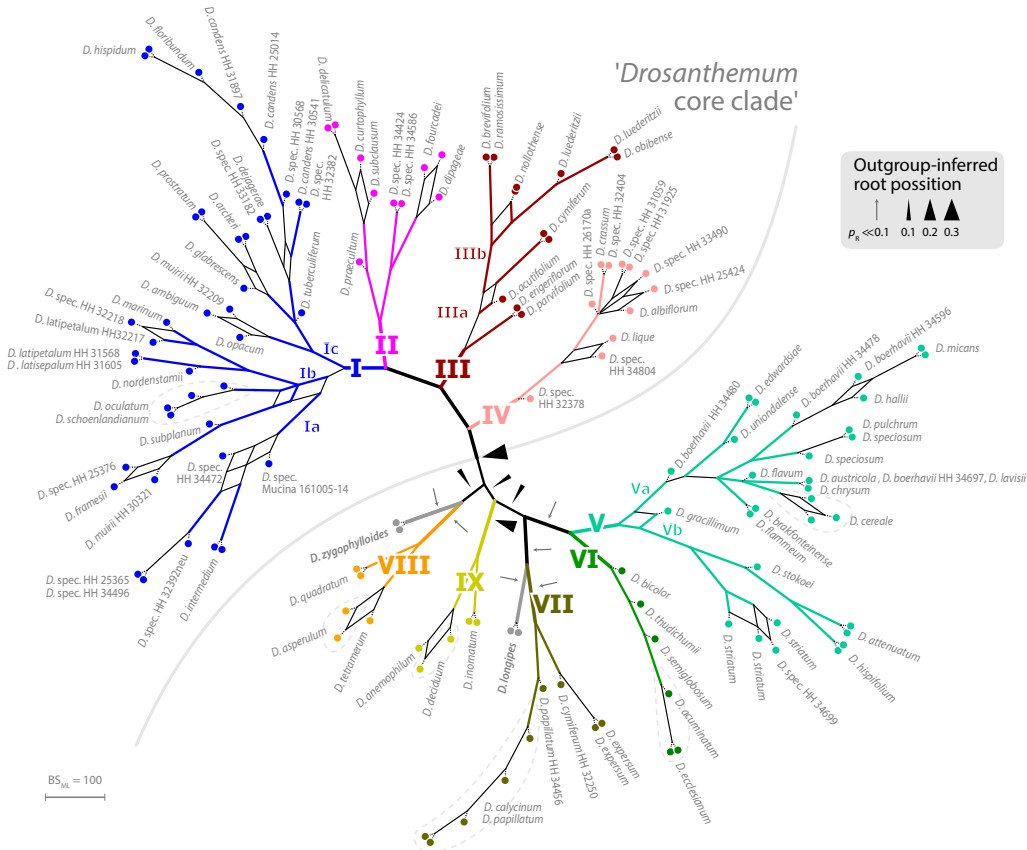

**Figure 3** **Bootstrap consensus network of *Drosanthemum*.** Consensus network based on 600 pseudoreplicate samples inferred by partitioned ML analysis of the cpDNA sequence data. Edge lengths are proportional to the frequency of the phylogenetic split in the pseudoreplicate sample. Branch colours and labels are as in Fig. 2. Black arrows indicate potential root positions inferred by outgroup-EPA, with arrow size proportional to the probability estimate $p_R$ (Supplemental Information 4, Table S4).

of ITS genotypes). Overall, the results obtained by outgroup-EPA are consistent with a fast radiation generating the main lineages early in the evolution of *Drosanthemum*.

## Inter- and intra-clade differentiation patterns

The haplotype analyses (of each gene region) is in overall congruence with the combined cpDNA tree (Fig. 2). However, in some genes and/or clades coherent mutational patterns are shared by several species, which lack uniquely shared sequence patterns in other gene regions. Thus, detailed haplotype networks (Figs. 4–7) further illuminate phylogenetic relationships in clades VII–IX (and the two isolated species *D. longipes* and *D. zygophylloides*), and further corroborate subgroups within clades I, III, and V (Figs. 2 and 3).

Clade I is divided into three subclades and the haplotype analysis supports a monophyly of clades Ia and Ic, but not Ib (Figs. 4A–4D). Members of clade Ib are characterized by haplotypes either ancestral to those found in clades Ia and Ic (*rpl16* intron, *trnK-rps16*) or unique and strongly divergent from each other (*rps16 -trn*Q). Clade II haplotypes are more

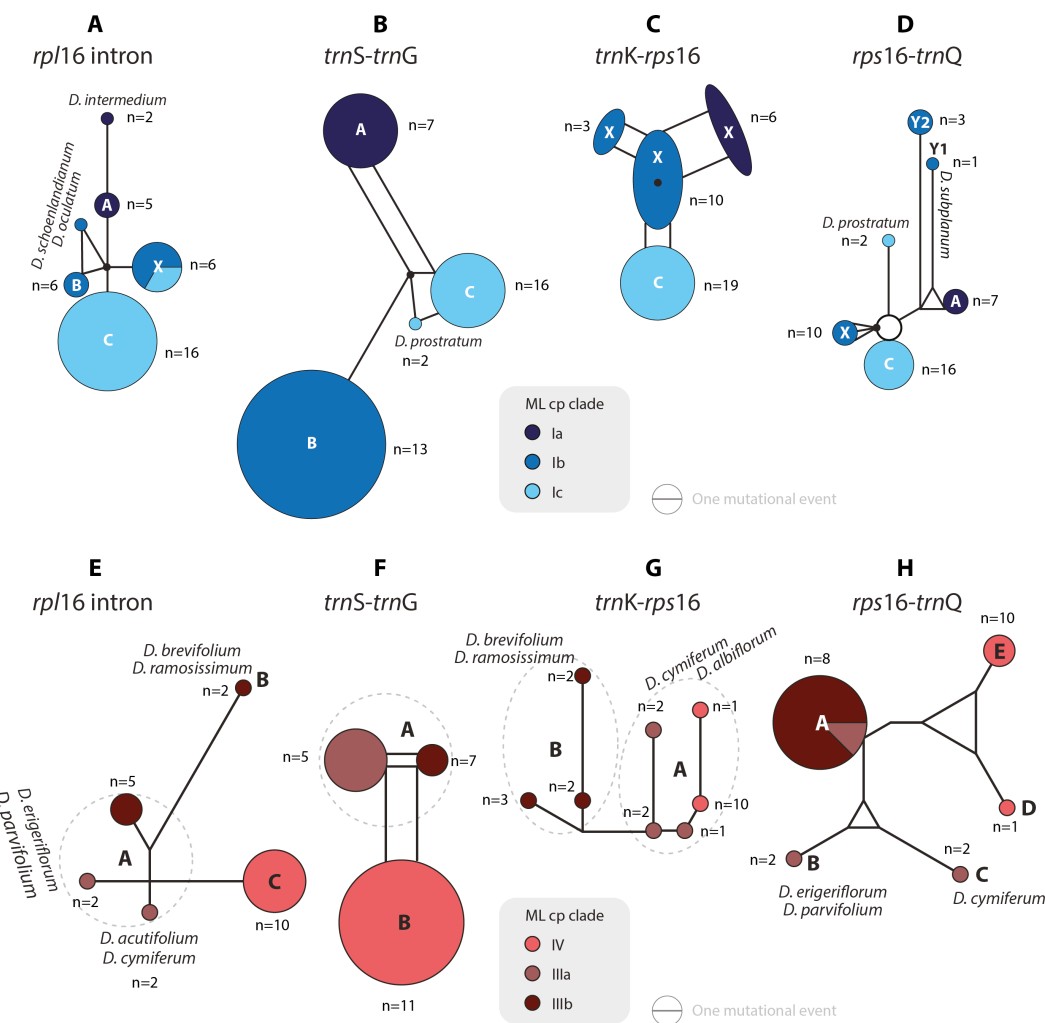

**Figure 4  Median-joining networks of *Drosanthemum* clades I–IV.** Collapsed Median networks; collapsed network portions (haplotype groups) represented by by circles; letters in bold refer to *Liede-Schumann et al. (2019)* (file Haplotyping.xlsx; archive includes full networks). Circle size does not show haplotype frequency but gives the maximum number of mutations between grouped haplotypes/ connective medians (a group's dimension); edge length (minimum) number of mutations between haplotype groups (*Grimm, 2019*). (A–D) Clade I. Note that subclade Ib is paraphyletic to clades Ia and Ic according to *rpl*16 intron, *trnK-rps* 16 and *rps*16-*trn*Q. Filled black circles (medians) denote position of the consensus sequence of the clade. (E–H). Clades III and IV. Note that grade IIIa (cf. Fig. 2) bridges between haplotype groups diagnostic for clades IIIb and IV, which could be an indication of paraphyly, i.e., grade IIIa species originate from a radiation predating the formation and subsequent radiation of clades IIIb and IV.

similar to ancestral haplotypes in clade I than to those in clades III or IV. The haplotypes in clades III and IV are very similar to each other (Figs. 4E–4H). Clade III is divided into a more diverse (likely paraphyletic) grade IIIa and a monophyletic clade IIIb (Figs. 2 and 3, 4E–4H). Within clade III, clade IIIb forms an increasingly derived (monophyletic) lineage (*D. luederitzii* + *D. obibense* → *D. nollothense* → *D. brevifolium* + *D. ramossissimum*) that starts with grade IIIa individuals having *D. cymiferum*-like morphology but being

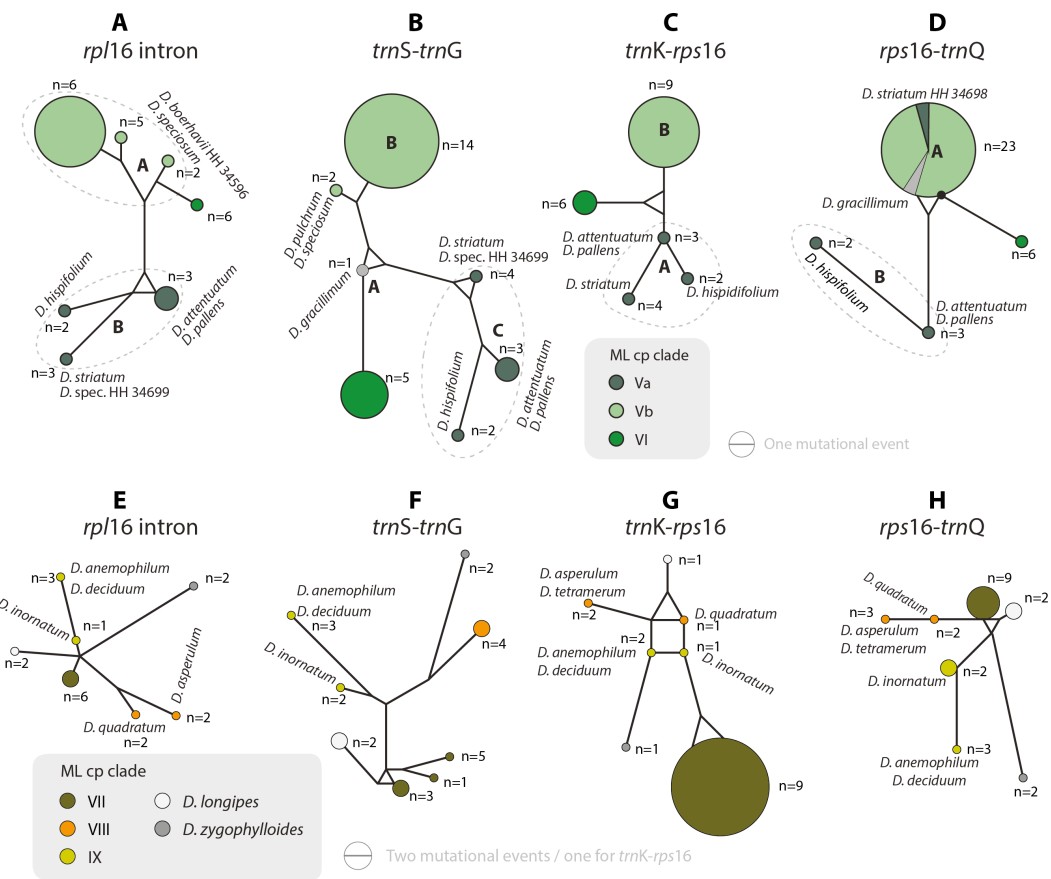

**Figure 5** **Median-joining networks of *Drosanthemum* clades V–IX.** Collapsed Median networks; collapsed network portions (haplotype groups) represented by circles; letters in bold refer to (*Liede-Schumann et al., 2019*) (file Haplotyping.xlsx; archive includes full networks). Circle size does not show haplotype frequency but gives the maximum number of mutations between grouped haplotypes/connective medians (a group's dimension); edge length (minimum) number of mutations between haplotype groups (*Grimm, 2019*). A–D. Clades V and VI. Note the central (*trn*S-*trn*G) or ancestral (*rps*16-*trn*Q) position of *D. gracillimum* (no *rpl*16 and *trn* K–*rps*16 data available). E–H. Clades VII–IX. Note that members of each clade are clearly differentiated but differ in the level of derivation per gene region.

genetically distinct from *D. cymiferum*. Clade V includes two sequentially coherent and mutually exclusive (reciprocally monophyletic) clades, Va and Vb (Fig. 3). In general, haplotypes of clade Vb show more unique shared mutational patterns than those of clade Va (Figs. 5A–5D). Figures 5A–5D also include the relatively similar haplotypes of the sister lineage, clade VI, which can be used to root the MJ networks (note that the edge length reflects the difference in the variable genetic patterns within clade V and does not include sequence patterns uniquely found in clade VI). Two markers, *trn*K-*rps*16 and *trn*S-*trn*G, reflect the potential reciprocal monophyly of both clades. *Drosanthemum gracillimum* is not included in clade Va or Vb (Figs. 2, 3). Only two of the considered cpDNA markers are available for this species, *trn*S-*trn*G and *rps*16-*trn*Q, with no lineage-diagnostic sequence
pattern and obviously showing the putative ancestral haplotype within clade V (Figs. 5A–5D).

Whereas haplotypes can be very divergent at the inter- and even intra-clade level (e.g., Fig. 4), they are relatively similar to each other in the smaller clades VII–IX (Figs. 5E–5H). *Drosanthemum longipes trn*S-*trn*G and *rps*16-*trn*Q haplotypes are highly similar to those of clade VII. Each gene region has a series of mutational patterns in which *D. longipes* and all members of clade VII are distinct from clade VIII and IX. In the lowest-divergent *trn*K-*rps*16 intergenic spacer region, the *D. longipes* haplotype can directly be derived from the one of clades VIII and IX (Figs. 5E–5H). *Drosanthemum longipes* is genetically closer to the putative *Drosanthemum* ancestor than to members of clade VII. In contrast, the haplotypes of *D. zygophylloides* are visibly unique within the genus (Figs. 5E–5H), which is also reflected in its long terminal branches in the cpDNA tree (Fig. 2).

### Identification of ITS genotypes in *Drosanthemum*

Analysis of nuclear ITS sequence variation reveals 62 genotypes, for which SP analysis produces an overall, but highly reticulated, star-shaped network with genotypes linked to various cpDNA lineages in the centre (Fig. 6; Supplemental Information S3). The least derived but most common genotypes are found in distantly related clades: genotype 33 in clade I and genotype 6 clade V (Fig. 6). Genotypes 6 and 33 resemble the consensus of all ITS genotypes differing only by a single point mutation (note that genotype 33 collects several subtypes differing in an indel pattern that is ignored by the SP network; Supplemental Information S3; for details see *Liede-Schumann et al., 2019*: files Haplotype.xlsx, DataSummary.xlsx). Genotype 3 is shared by members of clade VII and IX and is central to most other (including the most common 33 and 6; Fig. 6). Clades VII–IX and the two phylogenetically isolated species, *D. longipes, D. zygophylloides*, have unique, derived genotypes. The ITS genotypes of clades II, IIIb and IV can be derived from the most ancestral ones in clade I and IIIa. The fact that ITS evolution, a stepwise derivation of a putatively ancestral, consensus sequence into genotypes that are unique within clades and can be mapped on the cpDNA phylogeny (Fig. 2) indicates that ITS differentiation in the '*Drosanthemum* core clade' is in overall congruence with the cpDNA tree.

### Geographical clade structure in *Drosanthemum*

All clades retrieved in the present analysis show their own characteristic distribution pattern. All clades are present in the south-western Cape, and three clades (VI, VII, VIII) hardly extend beyond this narrow region. The species-rich clades I, II, and IV cover the largest areas. Clade V extends along the southern Cape coast and clades III and IX extend along the West coast (Fig. 7).

## DISCUSSION

### Genetic differentiation patterns indicate fast radiation initiating diversification within *Drosanthemum*

Phylogenetic analysis, in-depth haplotype analyses of cpDNA, and mapping of ITS evolution on the ML cpDNA tree point towards a rapid initial diversification within

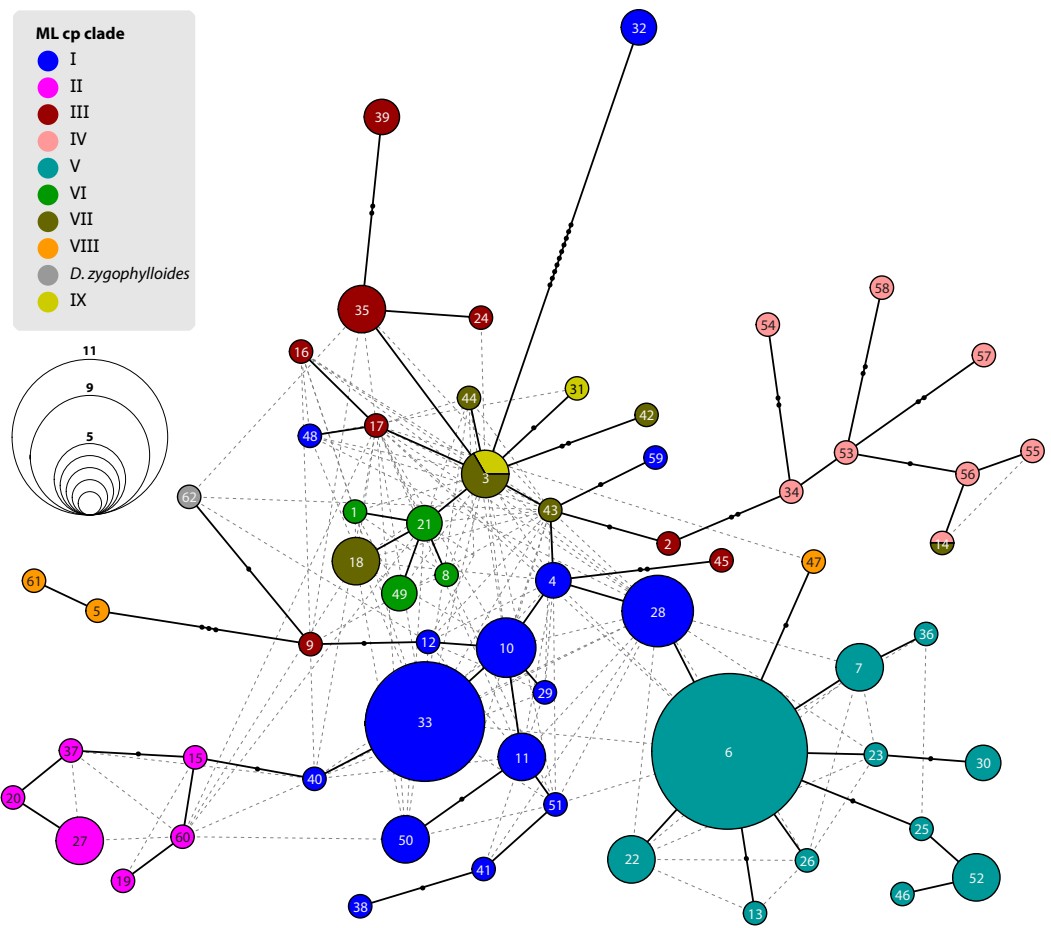

**Figure 6** **Statistical parsimony network of *Drosanthemum* ITS genotypes.** Network inferred by analysis of the ITS sequence data under an infinite site model. Genotypes are indicated by circles coloured according to clades inferred by cpDNA sequence analysis (see Figs. 3 and 4). Circle size indicate absolute frequency of genotypes (see legend). Black lines indicate steps in the network, filled black circles missing genotypes, and dashed grey lines alternative links. Genotypes in the centre of the graph are ancestral, those in the periphery most derived. Genotype 4 represents the genus consensus sequence found in several accessions of clade I (for details see Supplemental Information 3).

the genus *Drosanthemum*. The best outgroup-EPA inferred rooting position indicates *Drosanthemum* species to group in two large clades, with clade I–IV, the '*Drosanthemum* core clade', sister to clade V–IX (Fig. 2). The uncertainty in root position (Fig. 3; Supplemental Information S4) is consistent with a pattern expected in initial radiations (*Graham & Iles, 2009*; *Saarela et al., 2007*). The star-like (but reticulated) structure of the SP network (nuclear ITS data) suggests an initial bottleneck early in the evolution of *Drosanthemum* followed by rapid diversification (Fig. 6, Supplemental Information 3). Similarly, the plastid sequence variation provides sufficient information to resolve nine well-supported clades within the genus *Drosanthemum*. However, the 'backbone' relationships among the nine clades, or more specifically, the six lineages, are not resolved (Figs. 2 and 3). Taken together, the difficulties to separate and clarify the exact sequence of

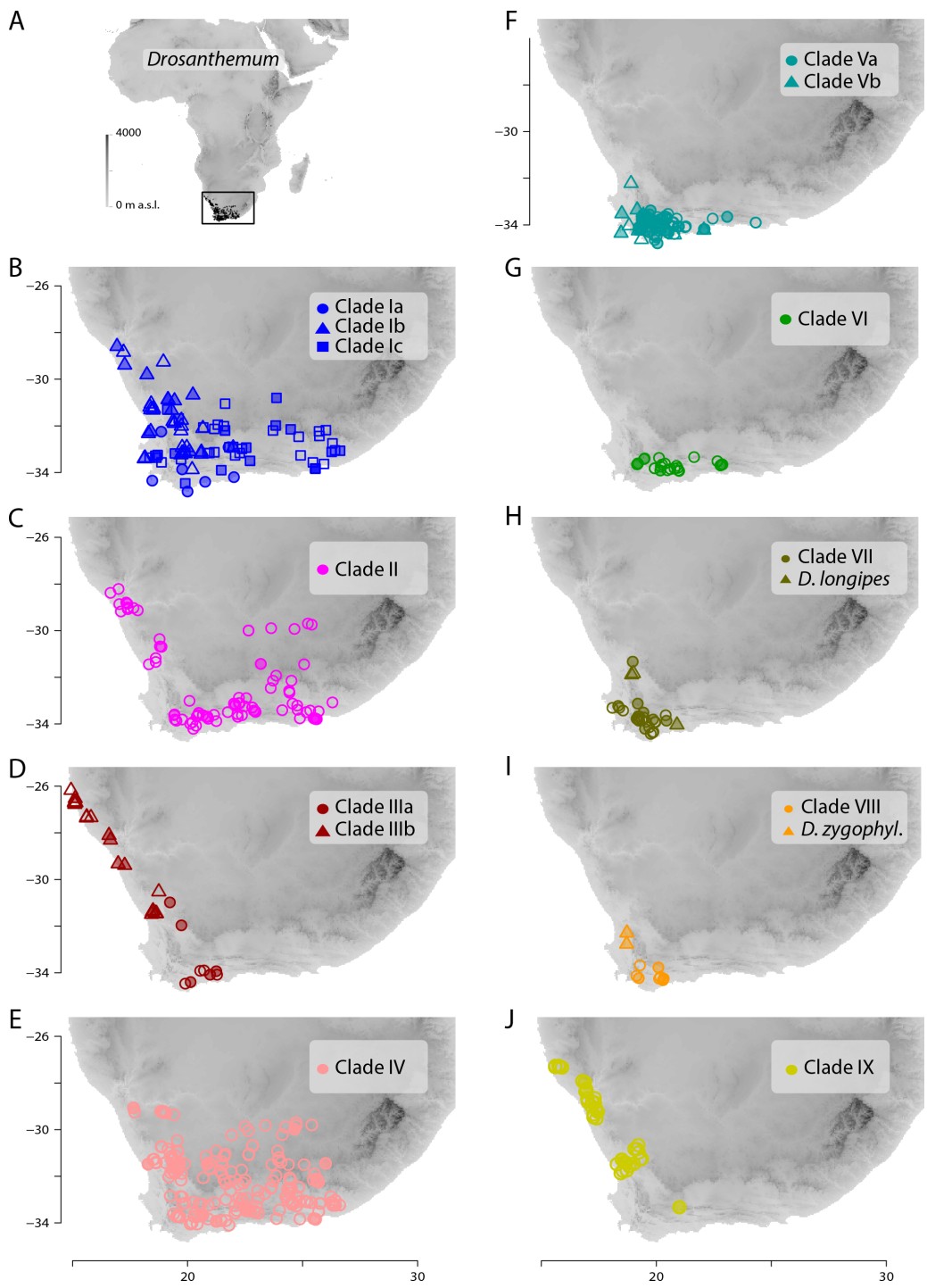

**Figure 7** **Distribution of *Drosanthemum*.** (A) Overall distribution of *Drosanthemum* in Africa. (B–J). Clade-wise distribution of *Drosanthemum* species in southern Africa. Filled symbols indicate accessions used in the phylogeny, empty symbols indicate the remaining accessions in the occurrence dataset of *Drosanthemum*. *D.zygophyl.*, *D.zygophylloides*. Maps were created using the elevation above sea level data from the WorldClim climate layers (*Hijmans et al., 2005*), with a spatial resolution of 30′ using the raster library v2.8-19 (*Hijmans, 2019*) in R v3.5.3 (*R Core Team, 2019*).

early branching events is a characteristic pattern in rapid evolutionary radiations among the plant tree of life, and has been found at various phylogenetic levels, for example, in Saxifragales (*Fishbein et al., 2001*), within the genus *Hypericum* (Hypericaceae; *Nürk et al., 2013*; *Nürk et al., 2015*) and in a group of South American *Lithospermum* (Boraginaceae; *Weigend et al., 2010*). It remains to be seen, however, whether analyses of nuclear markers apart from ITS support the patterns retrieved here.

## Phylogenetic inference reflects taxonomic classification

Within *Drosanthemum,* nine clades are revealed, which generally correspond to the recognized subgenera (*Hartmann, 2017a*), although some exceptions exist. The deviations in morphology-based classification and phylogenetic evidence produced in this study reveals cryptic species and several new relationships. For example, the species *D. zygophylloides*, *D. gracillimum*, and *D. longipes*, have either never been included into the subgeneric classification (*D. zygophylloides*), or phylogenetic evidence indicates affinities different from classification (*D. gracillimum*, *D. longipes*; Fig. 2). Considering our results, these species cannot be included in any of the proposed subgenera (*Hartmann, 2017a*). Note that both *D. longipes* and the species in clade IX shed leaves in summer and resprout with the winter rains.

Subgenus *Drosanthemum* is revealed as biphyletic, with most of its species in clade I, sister to clade II (subgenus *Xamera*; see below). *Drosanthemum hispidum*, the type species of *Drosanthemum*, groups in clade I (subclade Ic; Fig. 2). The rest of the species classified in subgenus *Drosanthemum* group within clade III. No morphological diagnostic characters are obvious to distinguish the clade III species from those in clade I, and thus, clade III is not yet circumscribed as a tenth subgenus. Likewise, subgenus *Drosanthemum* species in clade I group in three subclades Ia, Ib, and Ic, but morphological characters defining these clades cannot yet be named. Hence, this species-rich subgenus is obviously biphyletic, but species assigned to it are not distributed all over the tree, i.e., subgenus *Drosanthemum* does not appear to be a "dustbin" for species that cannot be assigned based on morphology to any other subgenera.

The discussed clades I–III, together with clade IV, constitute the informally named '*Drosanthemum* core clade'. Clade IV corresponds to the night-flowering subgenus *Vespertina* that is characterized by flowers of the long cone type (*Rust, Bruckmann & Hartmann, 2002*). Subgenus *Xamera* (clade II) is characterized by usually six-locular capsules and four tiny spinules below the capsule stalk and on older lateral branches (*Hartmann, 2007*). *Drosanthemum delicatulum* and *D. subclausum* of clade II also show this character, so that their listing under subgenus *Drosanthemum* in *Hartmann (2017a*: 508, 532) is clearly erroneous as is also indicated by the listing of *D. subclausum* among the species of *Xamera* in *Hartmann (2017a*: p 495). Conversely, *D. dejagerae* L.Bolus, attributed to *Xamera* by *Hartmann (2007)* and *Hartmann (2017a)* due to the presence of a six-locular capsules characteristic for the subgenus, is placed in clade Ic (subgenus *Drosanthemum* p.p.).

Of the six remaining subgenera, four—*Speciosa* (clade Va), *Ossicula* (clade Vb), *Necopina* (clade VI), and *Quastea* (clade VII)—group in one clade that is, however, not well supported

(Fig. 2) and also lacks obvious commonly shared, derived morphological characters. In particular, the stout and often large capsules (to one cm diam.) of subgenus *Speciosa* (*Hartmann & Bruckmann, 2000*) contrast strongly with the tender and smaller capsules of the other three subgenera. However, bone-shaped closing bodies in the capsules, considered unique for subgenus *Ossicula*, have also been found in capsules of *Speciosa* species (*Hartmann & Le Roux, 2011*), reducing their potential as a diagnostic character for *Ossicula*. This is illustrated by *D. austricola* L.Bolus, which is retrieved in subclade Va, corresponding to subgenus *Speciosa*, despite its conspicuous bone-shaped closing body, a character for which it was placed in *Ossicula* by *Hartmann (2008)*.

While the subgeneric classification of *Drosanthemum* (*Hartmann, 2007*; *Hartmann & Liede-Schumann, 2014*) is largely confirmed, a few unexpected placements of single species deserve mentioning. Of the three samples of *Drosanthemum cymiferum*, attributed to subgenus *Quastea* in *Hartmann (2007)*, only one sample was retrieved in the *Quastea* clade VII, the other two in clade III (*Drosanthemum* p.p.). This species was studied in some more detail in *Liede-Schumann, Meve & Grimm (2019)*, who did not find any consistent morphological differences between these samples and suggested a case of cryptic speciation (following the definition of *Bickford et al., 2007*). A similar case is found in *D. muirii* L.Bolus, of which the two samples are retrieved with good support in subclades Ia and Ic, respectively (Fig. 2).

**Distinct geographic distributions in the Greater Cape Floristic Region**

Inside the genus *Drosanthemum*, six lineages originate from a soft polytomy (precisely, they root in an unsupported part of the tree; Fig. 2), suggesting a radiation right at the start of the evolutionary history of *Drosanthemum*. To which extent this radiation was driven by ecological or geographical factors remains an open question. Interestingly, several clades comprising only 3–6 species are distributed over a restricted geographical range: clade VI *Necopina* (6 spp), clade VII *Quastea* (4 spp), and clade VIII *Quadrata* (3 spp) are restricted to the western part of the Cape Mountains (Figs. 7F–7I). One species-poor lineage, clade IX *Decidua* (3 spp.), extends along the West Coast into Namibia Fig. 7J). Species in clade V, 14 in clade Va *Speciosa* and 6 in clade Vb *Ossicula*, are almost restricted to the fynbos of GCFR (Fig. 7F), whereas the comparatively higher species number in *Speciosa* might be the result of more thorough studies in this showy, horticulturally valuable subgenus (e.g., *Hartmann, 2008*; *Hartmann & Le Roux, 2011*). Notably, these clades are genetically and morphologically coherent, that is, possess unique and derived sequence patterns as well as characteristic morphologies.

The more or less narrow distribution pattern of these clades (Figs. 7F–7I) contrasts to a wide distribution of the '*Drosanthemum* core clade' (Figs. 7B–7E), harbouring more widespread lineages with more species potentially indicating broader overall-habitat preferences: clade IV *Vespertina* (12 spp), clade II *Xamera* (8 spp) and the genetically and morphologically most diverse clade I (*Drosanthemum* p.p.; ≥ 55 spp; Fig. 7). The bulk of species diversity has been described in subgenus *Drosanthemum*, which falls in three subclades Ia–Ic not previously recognized (Fig. 2). These three subclades show distinct distribution patterns, with Ia restricted more or less to the fynbos area of GCFR, Ic

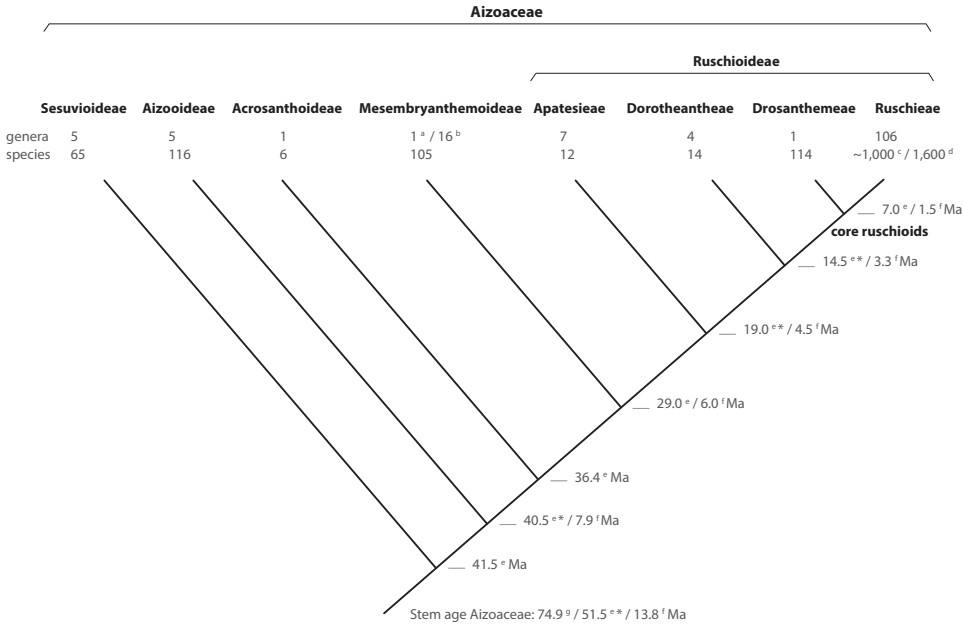

**Figure 8   Phylogeny of Aizoaceae.** A summary cladogram indicating recognized subfamilies (*sensu Klak, Hanáček & Bruyns, 2017a*) and tribes (*sensu Chesselet, Van Wyk & Smith, 2004*) detailing the number of genera and species and estimated node ages. Superscript letters denote reference: *a, Klak & Bruyns (2013)*; *b, Hartmann (2017a, 2017b)*; *c, Stevens (2001, onwards)*; *d, Klak, Bruyns & Hanáček (2013)*; *e, Klak, Hanáček & Bruyns (2017b)*; *f, Valente et al. (2014)*; *g, Magallón, Gómez-Acevedo & Sánchez-Reyes (2015)*. A superscript asterisk denotes ages according to *Klak, Hanáček & Bruyns (2017a*, Fig. S2).

extending far into the east and northeast, while Ib extends north to 28°S (Fig. 7B). Clade III, composed of species hitherto considered to belong to subgenus *Drosanthemum*, shows the most diverse distribution of all clades, with a southern group of poorly resolved species, and a lineage of several species extending to the northernmost locality of *Drosanthemum*, the Brandberg in Namibia (*Liede-Schumann, Meve & Grimm, 2019*; Fig. 7D).

Some more species-rich clades within *Drosanthemum* have also wide ecological preferences, with representatives both at lower and higher elevations. Morphological adaptations to arid habitats are capsules with deep pockets caused by false septa enabling seed retention (*Hartmann & Bruckmann, 2000*), which have been evolved in parallel in clade Ia and IIIb. However, whether the possession of false septa in the capsules is restricted to species of arid habitats remains an open question.

## CONCLUSIONS

In this study, we present a comprehensive phylogenetic investigation of *Drosanthemum,* a morphologically diverse genus that has so far been relatively overlooked in evolutionary studies of Aizoaceae. Our results confirm *Drosanthemum* (= Drosanthemeae) as sister lineage to Ruschieae, which is in accord with the 'core ruschioids' hypothesis (*Klak, Reeves & Hedderson, 2004*; *Klak, Bruyns & Hanáček, 2013*; Fig. 8). Additionally, our phylogenetic evidence signifies *Drosanthemum* as a genetically well-structured but heterogenous lineage

of mesomorphic plants that is, however, less species-rich than its sister clade; a pattern of diversity distribution common in the plant tree of life (*Donoghue & Sanderson, 2015*). Still, our analysis suggest that *Drosanthemum* is not simply a depauperate lineage sister to a radiation, but instead exemplifies a radiation by itself as indicated by complex plastid and nuclear DNA sequence differentiation patterns (Figs. 2, 3, 6), and the flower and fruit diversity present in the genus that is unusual for Aizoaceae.

Occurrence patterns among the evolutionary lineages might further indicate geographic factors playing a role in species diversification in *Drosanthemum*. While most of the evolutionary history of the genus seem to have taken place in a relatively mesic environment in the southwestern parts in the GCFR, several lineages apparently have started to adapt to more arid and/or winter-cold areas. Genetically relictual species from at least two early radiations co-exist among rapidly evolving lineages, reflecting species-delimitation problems in species-rich clades. This is mirrored in the present study that largely supports the current taxonomic concepts in *Drosanthemum* with few interesting exceptions, among others, cryptic species.

## ACKNOWLEDGEMENTS

Laco Mucina (Univ. of Western Australia) is thanked for a pleasant field trip and a sample of *D. zygophylloides* and Hans-Dieter Ihlenfeldt (Univ. Hamburg) for contributing several of the outgroup samples. SLS thanks the participants of the MSc Module F1 at the University of Bayreuth from 2008 to 2015 for their work on *Drosanthemum* herbarium specimens. Angelika Täuber and Margit Gebauer (UBT) are thanked for their enduring and conscientious lab work.

### Funding

This work was supported by two awards of the Mesemb Study Group (M.S.G.) in 2010 and 2013. The publication of this work was funded by the German Research Foundation (DFG) and the University of Bayreuth in the funding programme Open Access Publishing. The funders had no role in study design, data collection and analysis, decision to publish, or preparation of the manuscript.

### Grant Disclosures

The following grant information was disclosed by the authors:
Mesemb Study Group (M.S.G.).
German Research Foundation (DFG).
University of Bayreuth.

### Competing Interests

Alastair J. Potts is an Academic Editor for PeerJ.

## Author Contributions

- Sigrid Liede-Schumann conceived and designed the experiments, performed the experiments, analyzed the data, authored or reviewed drafts of the paper, and approved the final draft.
- Guido W. Grimm analyzed the data, prepared figures and/or tables, authored or reviewed drafts of the paper, and approved the final draft.
- Nicolai M. Nürk prepared figures, authored and reviewed drafts of the paper, and approved the final draft.
- Alastair J. Potts analyzed the data, authored or reviewed drafts of the paper, and approved the final draft.
- Ulrich Meve performed the experiments, analyzed the data, authored or reviewed drafts of the paper, curated herbarium material, and approved the final draft.
- Heidrun E.K. Hartmann conceived and designed the experiments, authored or reviewed drafts of the paper.

## DNA Deposition

The following information was supplied regarding the deposition of DNA sequences:

The sequences are available at ENA: LR030506 to LR030978.

## Data Availability

The data is available at Dryad: doi: https://doi.org/10.5061/dryad.n2z34tms2.

## Supplemental Information

Supplemental information for this article can be found online at http://dx.doi.org/10.7717/peerj.8999#supplemental-information.

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
