# Peer review of "Phylogenetic relationships in the southern African genus Drosanthemum (Ruschioideae, Aizoaceae)"

_PeerJ, doi:10.7717/peerj.8999_

## Round 0.1 · original submission · Major Revisions

Dear Dr. Liede-Schumann,

The reviewers think your manuscript may contributes to clarifies the evolution of Drosanthemum. Nevertheless, they have some concerns. In particular, a reviewer asks for a further analysis on cpDNA regions. Moreover, although reviewers find the text well written, they ask to clarify or simplify some parts.

So, I encourage you to improve the manuscript according to tips of reviewers. Please, respond point-to-point to the comments of reviewers to speed up the process of revision.

Once again, thank you for submitting your manuscript to PeerJ and we look forward to receiving your revision.

Sincerely,
Gabriele Casazza

Reviewer 1 ·

Basic reporting

The writing of the manuscript is vague, with many grammar issues. I am listing a few examples below. Please check the entire manuscript carefully. In addition, in general the flow of the MS can be improved by using topic sentences to guide readers.

Line 27: “73 species represented by multiple accessions of Drosanthemum” is vague. Do you mean “73 species of Drosanthemum with multiple accessions per species”?

Line 28: “phylogenetically studied” is awkward.

Line 29: It is unclear what does it mean by “phylogenetic affinities not obvious in the phylogenetic tree”

Line 37–38: “molecular and morphologically diverse” is grammatically incorrect. Also “molecular diverse” is vague and should be reworded.

Line 120: “are nested in”, instead of “group inside”

Line 130: “More generally”, instead of “More general”

Line 281 to Line 324: paragraph very long. Break into multiple paragraphs and use topic sentences to guide readers.

Line 326: “Genetic diversity patterns” is vague. How about something like “Alignment matrices”?

Line 364: “relationships among the six lineages were weakly supported” instead of “without supported relationships”.

Line 453–459: very long sentence. Shorten or break into multiple sentences. In addition, reticulation vs. incomplete lineage sorting are two distinctive processes. They are not “consistent with” each other.

Fig. 3: specify that outgroups are removed in the figure legend, and add “Subg. ” to each of the names on the right to avoid confusion

Experimental design

In general, the sampling, marker used, and methods are appropriate for the research questions. My only minor comment is that the two cpDNA regions should be analyzed separately before concatenating them.

Validity of the findings

In general, the results are valid. Given the low variation in ITS, the phylogenetic analyses mainly relied on two cpDNA regions. It is unclear whether analysis of additional nuclear regions would recover the same phylogenetic relationship. This caveat should be explicitly addressed in Discussion.

Additional comments

Given the main focus of the MS on the phylogenetic relationships within Drosanthemum, the introduction is too long. It is unclear how this long list of character evolution and phylogenetic relationships in Aizoaceae are relevant to Drosanthemum. I would suggest the authors significantly shorten the introduction and focus on how evolution of a subset of characters relate to the evolutionary context of Drosanthemum, and remove the statement in Line 54 on “a synoptic overview on the current classification of the family”.

Reviewer 2 ·

Basic reporting

According to the authors, the raw data is available in the Dryad repository, but I could not check this as the link they provide was not working, I guess this is due to an embargo.

Experimental design

No comment

Validity of the findings

No comment

Additional comments

In this manuscript,the authors present a phylogenetic study of Drosanthemum with a wide taxonomic sampling (more than 60% of the genus) that clarifies the evolution of this genus. Previous studies had just focused on very few species (around 10% of the genus). The authors have conducted a thorough work to insure that specimens sequenced are well-identified, an important point givent that it is a challenging taxonomic group. Another strength of the study is that the authors included several individuals for species that are more morphologically variable and widespread.
The research questions of the study are well defined and are relevant to understand the evolution of the genus studied. The methods used and the data analysed are appropriate to fulfill the main aim of the study, and the discussion and conclusions derived from their results are sound.
The article is clearly written, the introduction provides sufficient background and is well referenced. The methods and results are clearly detailed. The discussion and conclusions are well stated.

Concerning to weaknesses of the study, I only detected minor text issues that I detail below:

-The authors speak of "pseudo-cryptic speciation", but following Bickford et al 2007 (https://doi.org/10.1016/j.tree.2006.11.004) I think it is more appropriate to speak of cryptic speciation, as it does not necessarily imply a sister relationship, among others (see the section "What are cryptic species?" of that article). In relation to this, I think that the reference to Sáez et al. 2003 (a study on coccolithophores) that the authors included when speaking of pseudo-cryptic speciation is not relevant here.

-l. 157 "extra-ordinary" should be "extraordinary"
-l. 183 replace "or" by "and", more appropriate
-l.399 typo: "haplotyp"
-l. 512: typo: "is largely confirmed a few ..." should be "is largely confirmed, a few..."
-l. 527: should say "are restricted" instead of "restricted"
-l.528: should say "extends" instead of "extend"
-l.530: should say "Ossicula are" instead of "Ossicula, are"
-l.565: rephrase the end of this sentence in a more clear way: "the, for Aizoaceae unusual, flower and fruit diversity present in the genus."
-l.573: "this is mirrored in in the" should say "this is mirrored in the"

Reviewer 3 ·

Basic reporting

See general comments.

Experimental design

See general comments.

Validity of the findings

See general comments.

Additional comments

Review of Phylogenetic relationships in the southern African genus Drosanthemum (Ruschioideae, Aizoaceae).
This manuscript presents an interesting phylogenetic study of an interesting genus of Aizoaceae. For the first time, authors generated a data set with good phylogenetic representation of the the genus (73 of 114 species included), based on highly informative cpDNA markers and not so informative ITS data. Moreover, they compiled an occurrence data set. With these, they reconstruct a phylogeny, but this is not formally analysed (i.e., not dealt with in "Results"). Overall, I find the study interesting and relevant, but I have some concerns regarding the presentation, especially of the ITS data.

The whole paper is rather long, and although it is reasonably well written, it includes a fair amount of unusual phrases, such as "branching artifacts" (l 243), "pseudo-cryptic species" (l 574), "biphyletic" (l 480) "molecularly homogeneous lineage" (l 177). Although these terms can be understood with some good will, they hamper the flow of the paper.

The introduction is quite long, but contains a fairly complete overview of the phylogenetic relations within the relevant parts of the family, so that it contributes to the paper.

The questions at the end of the introduction are eseentially valid, but phrased unnecessarily complex.
Q1, l177-179, is formulated in an unusual phrasing and unnecessarily vorbose, "whether Drosanthemum constitutes a coherent, molecularly homogeneous lineage sister to Ruschiaea, or whether it forms an agglomeration (or grade) of several mesomorphic lineages", while the question is plainly and simply "Is Drosanthemum a monophyletic lineage sister to Ruschiaea?". That's a fine question that doesn't need other wording. Similarly, for Q2, I would drop the words "covering all species diversity" from line 180. Simply ask whether the morphologically delineated subgenera are monophyletic.
Q 3 in essence is two questions: is subgenus Drosanthemum also monophyletic, and, does subgenus Drosanthemum contain cryptic species. I don't see how the first aspect of Q3 is fundamentally different from Q2, as both questions deal with the circumscription of subgenera. Therefore, you should rephrase to reflect the second aspect, namely, whether multiple accessions within species within the dustbin subgenus Drosanthemum are eachother's closest relatives. Q4 is fine as formulated, though a formal analysis addressing this question is lacking. If you want to do this, you could, for instance, analyse the pairwise distances between all accessions, and analyse this with a PCoA.

The methods are logically structured and complete, but they could be shortened. In various places you present a fairly complete list of what you did, and then again explain the same information later (e.g., l 240-253 is largely superfluous). For instance, l 243 you defer dealing with "branching" artifacts to the next paragraph, but there you barely deal with it and in stead refer to SI and literature. Perhaps you can streamline all this.

The Dryad link did not yet work.

l 246-248. This is an interesting application of the EPA algorithm that I have not yet seen. The algorithm was not designed for this application (though I suspect it should work). If you know of other studies to use it to find a root position, please cite them.
I don't really understand, however, why this approach is preferred over simply including the outgroup DNA data in a phylogenetic analysis. Does in any way it leverage information not available in a normal RAxML analysis? Apparently, they avoid what you call "ingroup outgroup branching artifacts", but you don't explain what they are, how they arise, you only refer to the SI of *another* paper for examples, so this remains unclear. Certainly, both EPA and a "normal" raxml run are based on computing the likelihood of alternative topologies so results must be very similar.

ITS data. I think you try to make too much of this essentially not very informative marker. I think it suffices to state that including it with the cpDNA data did not alter the topology and analysing on its own did not reveal well-supported conflict with the cpDNA results. Including it within Figure 3 is very distracting and should be avoided. The cpDNA results and outgroup placement analyses indicate that the branch to the ingroup is too long to confidently place the outgroup. Hence, the cpDNA evolves too fast to be informative. The ITS data, however, evolves too slow to infer relations within the outgroup, but that may suggest it is a useful marker to place the outgroup?

Median joining networks. These results, in my opinion, add very little information to the paper. Mostly, they corroborate the cpDNA phylogenetic results. Therefore, I would not include them in the main paper, they are distracting.

I'm a little disappointed that you could not make more of the distributional patterns, except discussing them briefly. If you want to keep them, you should include a sentence or two in the results. The clades are geographically quite highly structured, but for readers not familiar with the Cape region, perhaps you can map them onto a map defining major climate types, to corroborate your discussion.

The discussion overall is interesting and reflects the intimate knowledge of the authors of the group of study.

Figures.
10 Figures is certainly a lot for this paper and in my opinion, it dilutes the valuable message of the paper unnecessarily.

Figure 1 is a a systematic classification, not the result of a phylogenetic analysis based on the data you collected. Therefore, it is more appropriate to present it as a table. Alternatively, since your results underline this figure, you can include it as the "summary", but then it should be the last figure

Figure 2 is nice.

Figure 3 is a bit confusing. It deals with the cpDNA, but it isnt clear what "information from ITS data" means. This is the "main tree figure". Can't you simply do a parsimony network of the ITS data and demonstrate that it doesnt in major ways contradict the cpDNA tree?

Figure 4 is nice.

The network presented in Fig 5-8. is a bit unconventional and therefore confusing. In particular, the circles are not proportional to the number of individuals (as is typical in a haplotype network) but to sequence divergence within clades. Overall, I don't really see the point in the light of the rest of the results. If you want to keep these figures, figs 5-8 should be collapsed into a single figure.
The reason that I don't think these figures are very important is that all they signify is that there are not a large number of mutations in the data. Therefore, you quickly form networks.

---

## Round 0.2 · Minor Revisions

Dear Dr. Liede-Schumann,
the reviewers think your manuscript was strongly improved and recommend the publication of the manuscript. One reviewer has a couple of minor remarks. So, I ask you to make these few revisions before the manuscript will be accepted for publication. Once again, thank you for submitting your manuscript to PeerJ and we look forward to receiving your revision.
Sincerely,
Gabriele Casazza

Reviewer 1 ·

Basic reporting

No additional comments.

Experimental design

No additional comments.

Validity of the findings

No additional comments.

Additional comments

The authors did a nice job improving the manuscript. I only have a few very minor wording suggestions.

Line 84: “First-branching” is vague. How about “sister to the rest of Aizoaceae”?

Line 243: Do you mean “partitioning” instead of “portioning”?

Line 336: “among” instead of “between” the six lineages.

Line 436: Given potential concerted evolution in the ITS region, and there is no real test for the population size, it is unclear that there is support for an “initial bottleneck”. Perhaps remove this as it is not essential to the main questions of this MS.

Table 1: without some grid lines it is not very intuitive what are included in “Mesembs” and which “core ruschioids"

Reviewer 2 ·

Basic reporting

No comment

Experimental design

No comment

Validity of the findings

No comment

Additional comments

The authors have successfully responded to the (minor) concerns of my previous review. I therefore recommend the publication of the manuscript.

Reviewer 3 ·

Basic reporting

no additional comments

Experimental design

no additional comments

Validity of the findings

no additional comments

Additional comments

The authors have provided a careful revision of the paper, where they incorporated many of the points I previously raised. In particular, the readability of the paper is considerably improved by rephrasing the questions at the end of the introduction, and restructuring elsewhere. The authors did not agree with me on all fronts, but the parts where our opinions diverge are largely matters of presenting results, rather than their validity.

A few minor remarks:
- I don't intuitively understand the rank of the informal names "Mesembs" and "core ruschioids" in Table 1. It would be helpful to point this out, e.g. with footnotes.

Fig 2. In the legend, "Edge lengths are scaled on expected number of substitutions" should probably read "Edge lengths are drawn proportional to the expected number of substitutions per site".

---

## Round 0.3 · accepted · Accept

Dear Dr. Liede-Schumann,

I am very pleased to say that your paper "Phylogenetic relationships in the southern African genus Drosanthemum (Ruschioideae, Aizoaceae)" is accepted for publication in PeerJ. Congratulations!

Thank you for submitting your work to PeerJ.

Sincerely,

Gabriele Casazza